# Ultra-Small and Metabolizable Near-Infrared Au/Gd Nanoclusters for Targeted FL/MRI Imaging and Cancer Theranostics

**DOI:** 10.3390/bios12080558

**Published:** 2022-07-24

**Authors:** Xiawei Dong, Jing Ye, Yihan Wang, Hongjie Xiong, Hui Jiang, Hongbing Lu, Xiaohui Liu, Xuemei Wang

**Affiliations:** 1State Key Laboratory of Bioelectronics (Chien-Shiung Wu Laboratory), School of Biological Science and Medical Engineering, Southeast University, Nanjing 210096, China; dongxwseu@163.com (X.D.); jing832808@163.com (J.Y.); yihanwangxynu@163.com (Y.W.); 18056356092@163.com (H.X.); 101010998@seu.edu.cn (H.J.); 2Department of Biomedical Engineering/Computer Application, Fourth Military Medical University, Xi’an 710032, China; luhb@fmmu.edu.cn

**Keywords:** Au/Gd@FA NCs, dual-mode probe, FL/MRI imaging, reactive oxygen species (ROS), cancer theranostics

## Abstract

Tumor accurate imaging can effectively guide tumor resection and accurate follow-up targeted therapy. The development of imaging-stable, safe, and metabolizable contrast agents is key to accurate tumor imaging. Herein, ultra-small and metabolizable dual-mode imaging probe Au/Gd@FA NCs is rationally engineered by a simple hydrothermal method to achieve accurate FL/MRI imaging of tumors. The probes exhibit ultra-small size (2.5–3.0 nm), near-infrared fluorescence (690 nm), high quantum yield (4.4%), and a better T_1_ nuclear magnetic signal compared to commercial MRI contrast agents. By modifying the folic acid (FA) molecules, the uptake and targeting of the probes are effectively improved, enabling specific fluorescence imaging of breast cancer. Au/Gd@FA NCs with good biosafety were found to be excreted in the feces after imaging without affecting the normal physiological metabolism of mice. Intracellular reactive oxygen species (ROS) increased significantly after incubation of Au/Gd@FA NCs with tumor cells under 660 nm laser irradiation, indicating that Au/Gd@FA NCs can promote intracellular ROS production and effectively induce cell apoptosis. Thus, metabolizable Au/Gd@FA NCs provide a potential candidate probe for multimodal imaging and tumor diagnosis in clinical basic research. Meanwhile, Au/Gd@FA NCs mediated excessive intracellular production of ROS that could help promote tumor cell death.

## 1. Introduction

The rapid development of nanomedicine has provided new solutions for the diagnosis and treatment of diseases. In particular, the construction of multi-functional nanoprobes can meet the needs of multiple functions at the same time and realize applications in biomedical fields, such as multi-mode imaging, integration of diagnosis and treatment, visual monitoring, and others [1,2,3]. Multimodal imaging [4,5,6,7] can effectively overcome the inherent limitations of a single modality, such as fluorescence (FL) imaging, magnetic resonance imaging (MRI), and computed tomography (CT), and organically combine two or more imaging modalities to meet the requirements of accurate and fast imaging in clinical diagnosis. FL/MRI imaging [8,9,10] can not only improve the imaging detection depth and spatial resolution by using MRI but also greatly improve the sensitivity of FL imaging, which can provide relevant information at the cellular level and even guide surgical resection of the tumors in clinical practice, which is a dual-modal imaging technology with broad clinical application prospects. Currently, the FL/MRI dual-mode molecular probes provide two signals simultaneously by organically combining gold-based nanomaterials [11,12], carbon-based nanoparticles [13,14], upconversion nanoparticles [15,16], iron oxide/iron trioxide nanoparticles [17,18], gadolinium chelates [19,20], and other materials. These near-infrared fluorescent probes [21,22] can effectively avoid the interference of the autofluorescence of organisms and greatly improve the detection depth (up to several centimeters), which is more promising for biomedical and clinical applications. Compared with other near-infrared fluorescent probes, gold-based nanoparticles are widely used as probes due to their good biocompatibility, controllable morphology and size, surface modification, and stable fluorescence properties.

Compared with iron-based contrast agents and small molecule probes, paramagnetic MRI contrast agents containing Gd [23,24] can alter the relaxation time of protons and shorten longitudinal relaxation times (T1) and are widely used in clinical and scientific studies. However, due to the high toxicity of free Gd(III), the formation of derivatives by chelation is usually required prior to use. For example, DeRosa′s group [25] prepared a chelate of the fibrinogen inducer Gd(III) (Gd(III)-NOTA-FA), specifically targeting the location and accumulation of blood clots, and successfully implemented MRI imaging of blood clots. The common methods of binding to fluorescent materials are through shell, conjugation, doping, and different combination methods that allow the probes to present different structures and uses. For example, Liu′s group [26] prepared Au@Prussian blue-Gd@ovalbumin nanoparticles (APG@OVA. NPs), with a core-shell structure, which was used to track and activate dendritic cells (DCS).

However, the biggest drawback of FL/MRI dual-mode probes is their large size and the need to improve penetration and safe metabolism within the tumor. In addition, probes have a very limited role in intra-tumor cell aggregation using passive targeting, such as the enhanced permeability and retention (EPR) effect, is very limited. To enhance the targeting delivery of the imaging probes, surface modification of the probe is required, such as modified monoclonal antibodies [27,28,29], aptamers [30,31,32], and biological small molecules [33,34], to effectively improve the efficiency of cell internalization. Therefore, there is an urgent need to develop FL/MRI dual-mode probes that are small in size, highly targetable, and biosafe.

Herein, we prepared Au/Gd@BSA NCs by a simple hydrothermal method, modified folic acid molecules with tumor-targeting effects, and successfully synthesized Au/Gd@FA NCs dual-mode imaging probes for FL/MRI dual-mode targeting imaging of breast cancer cells and tissues (Figure 1). The prepared Au/Gd@FA NCs have a small size (2.5–3.0 nm), near-infrared fluorescence (690 nm), high quantum yield (4.4%), targeting ability, and good biocompatibility. In addition, due to the doping of Gd elements, Au/Gd@FA NCs have the ability of T_1_-nuclear magnetic signal and near-infrared fluorescence, as well as high fluorescence quantum yield, which successfully realized FL/MRI dual-mode targeted imaging for in vivo bioimaging of breast cancer. In addition, Au/Gd@FA NCs have good biosafety and can be excreted with feces after imaging without affecting the normal physiological metabolism of mice. More importantly, intracellular ROS increased significantly by confocal imaging after incubation of Au/Gd@FA NCs with tumor cells under 660 nm laser irradiation, indicating that Au/Gd@FA NCs can promote intracellular ROS production and effectively induce cell apoptosis. Thus, ultra-small and metabolizable near-infrared Au/Gd@FA NCs provide a potential candidate probe for clinical multimodal imaging and cancer treatment.

## 2. Materials and Methods

### 2.1. Materials

Gd(NO_3_)_3_·6H_2_O, BSA, EDC, NHS, and FA were purchased from Aladdin. Gold chloride acid, sodium hydroxide, 3-(4, 5-dimethyl–2-thiazolyl)-2, and 5-diphenyl-2–H-tetrazolium bromide (MTT) were purchased from Chinese medicine. A DCFH-DA (2′, 7′-dichlorofluorescein diacetate) reactive oxygen species assay kit was purchased from Sangon Biotech (Shanghai, China). YO-PRO-1/PI Apoptosis Kit was purchased from Beyotime Biotech. DMEM and fetal bovine serum (FBS) were purchased from GIBCO. The rest of the reagents not mentioned were purchased from Aladdin. All reagents were used directly, without any treatment. Deionized (DI) water was used in the polymerization recipe.

### 2.2. Preparation of Au/Gd@BSA NCs 

The preparation of Au/Gd@BSA NCs was improved based on previous literature reports. To put it simply, 50 mM HAuCl_4_ and BSA aqueous solution were evenly mixed at 37 °C, then a certain volume of 50 mM Gd(NO_3_)_3_ solution, adjusted pH to 8 with NaOH solution overnight. The prepared probe was dialyzed with PBS (pH = 7.2) for 24 h and then freeze dried.

### 2.3. Preparation of Au/Gd@FA NCs 

Twenty milligrams of Au/Gd@BSA NCs was dissolved in 2 mL PBS, 30 mg EDC/NHS (m:m = 1:2) was added, and 10 μL 1 × 10^−4^ mol/L FA solution was added, and the reaction was in the dark for 2 h. The precipitate was then collected by centrifugation, washed three times with ultrapure water, and stored at 4 °C.

### 2.4. Characterization of Au/Gd@FA NCs 

Transmission electron microscopy (TEM) (Hitachi, Tokyo, Japan, JEM2100) was used to study the size and morphology of Au/Gd@FA NCs. UV-vis spectra (Shimadzu, Kyoto, Japan, UV2450) and fluorescence spectra (Shimadzu, RF530C) were recorded using a spectrophotometer. Quantum yield (QY) was recorded by fluorescent spectrophotometer (Edinburgh, UK, FLS920). X-ray photoelectron spectra (XPS, United States, PHI-5000C), energy-dispersive X-ray spectroscopy (EDS, Zeiss, Oberkochen, Germany, ultra plus), and Fourier infrared spectrometer (FT-IR, Thermo Scientific, Waltham, MA, USA, Nicolet iS10) were used to characterize the chemical composition. The element content was collected using an inductively coupled plasma source mass spectrometer (ICP-MS, Perkin Elmer, Waltham, MA, USA, NexlON1000G). Confocal images were obtained using a confocal laser scanning microscope (Leica, Wetzlar, Germany, TCS SP8). Small animal imaging system in vivo (Perkin Elmer, IVIS Lumina XRMS Series III).

### 2.5. MTT Assay 

The cytotoxicity of Au/Gd@FA NCs was evaluated by MTT assay. Mouse Mammary tumor cells (4T1 cells) and normal human hepatocytes (L02 cells) were cultured in complete culture medium until the logarithmic growth phase, inoculated on 96-well plates, and incubated overnight. Different concentrations of Au/Gd@FA NCs were added and incubated for 24 h. Add 20 μL (1 mg/mL) MTT into culture medium for 4 h, discard supernatant, add 150 μL DMSO (5 mg/mL) and shake horizontally for 10 min. The absorbance of each well at 570 nm was measured, and the cell survival rate was calculated. The laser irradiation group was irradiated at 660 nm (1.2 W/cm^2^) for 6 min. The control group was not exposed to a laser. An electronic laboratory notebook was not used. The formula is as follows:Cell viability (%)=[ODexperiment−ODblank][ODcontrol−ODblank]

### 2.6. Laser Confocal Fluorescence Imaging

4T1 cells (6 × 10^4^ cells/well) and L02 cells (6 × 10^4^ cells/well) in the logarithmic growth phase were inoculated into confocal cell culture dishes, and 3 mL DMEM was added, then placed in a moist incubator overnight. The newly prepared Au/Gd@FA NCs (60 μg/mL) and Au/Gd@BSA NCs (60 μg/mL) were added to the cells for 6 h, respectively. The culture medium was discarded, and the cells were immersed in PBS several times after washing with PBS. FL data were collected by confocal laser microscopy (Leica TCS SP5). The control group was given a PBS solution. Confocal 3D scanning was performed on tumor cells co-incubated with Au/Gd@FA NCs (60 μg/mL) to observe the distribution of Au/Gd@FA NCs in the cells.

### 2.7. Relaxometry and In Vitro MRI 

Au/Gd@FA NCs and Gd-diethylenetriamine pentaacetic gadolinium (Gd-DTPA) solutions with varying concentrations were prepared for T_1_ relaxivity measurements and T_1_-weighted MRI at 25 °C by 1.5 T MRI scanner (Bruker Biospin, Rheinstetten, Germany). The relaxivity values of *r*_1_ were calculated by fitting the 1/T_1_ relaxation time (s^−1^) versus the Gd concentration (mM) curves.

### 2.8. In Vivo FL Imaging and MRI Analysis 

All animal experiments were approved by the National Institute of Biological Science and Animal Care Research Advisory Committee of Southeast University. All laboratory operations followed the guidelines of the Southeast University Animal Research Ethics Committee. The 3-week-old female BALB/c nude mice were selected as model animals and purchased from the Animal Experiment Center of Yangzhou University. 4T1 cells at the logarithmic growth stage were suspended in serum-free RPMI-1640 medium. One hundred microliters was taken and inoculated subcutaneously in the armpit of nude mice for 1–2 weeks until solid tumors grew to about 0.5–1 mm^3^. 

#### 2.8.1. In vivo FL Imaging

Au/Gd@FA NCs solution (100 μg/mL, 200 μL) was injected into the tail vein. At 0, 2, 4, 6, 8, 10, 12, 24, and 36 h after injection, the data of nude mice were collected and analyzed using an in vivo imaging system. The parameters were set as excitation wavelength 460–480 nm and emission wavelength 650–700 nm. The average fluorescence intensity of the tumor site was semi-quantitatively analyzed using imaging software. The nude mice were euthanized 8 h and 36 h after injection, respectively, and the tumor and main organs (heart, liver, spleen, lung, kidney, cerebrum, and intestine) were dissected for FL imaging and analysis ex vivo.

#### 2.8.2. In Vivo MRI

MR images were obtained by injection of Au/Gd@FA NCs solution (200 μL, 100 μg/mL) into caudal vein for 8 h. MRI was performed using a 7.0 T MRI scanner (Bruker Biospin, Germany) using T_1_-weighted sequences (Mapping, TR/TE = 6.83/2.26 ms, thickness = 2 mm).

### 2.9. Intracellular ROS Detection 

A DCFH-DA reactive oxygen species assay kit was used to detect total intracellular ROS. In a laser confocal petri dish, 4T1 cells were inoculated, and newly configured Au/Gd@FA NCs (60 μg/mL) were added and incubated in a cell incubator for 8 h. One group was irradiated with a 660 nm (1.2 W/cm^2^) laser for 6 min, and the other group was not irradiated with a laser. Discard the medium carefully, and wash with PBS. Each group was given DCFH-DA (10 mmol/L, 1 μL), and incubated for 20 min. Excess DCFH-DA was removed with PBS washing. Imaging was performed with a confocal laser scanning microscope (λex = 488 nm, λem = 500–560 nm).

Apoptotic cell staining: The apoptosis of tumor cells after laser irradiation was detected by YO-PRO-1/PI apoptosis kit. At a logarithmic growth stage, 4T1 cells were inoculated in a confocal culture dish for 24 h, and then cultured for 8 h with Au/Gd@FA NCs (60 μg/mL). The experimental group was given 660 nm (1.2 W/cm^2^) laser irradiation for 6 min, and the control group was not given laser irradiation. Cells were stained for 30 min according to the instructions of YO-PRO-1/PI apoptosis kit, and cell apoptosis was observed under a confocal fluorescence microscope.

## 3. Results and Discussion

### 3.1. Synthesis and Characterization of Au/Gd@FA NCs

As shown in Figure 1, Gd^3+^ was added into the reaction system by doping, and Au/Gd@BSA NCs were prepared by a one-step template method synthesis [35]. Subsequently, Au/Gd@FA NCs were obtained by covalently linking FA on the surface. By changing the volume of Gd(NO_3_)_3_ and adjusting the molar ratio of HAuCl4 to Gd(NO_3_)_3_ (1:1, 1:1.5, and 1:2), three kinds of Au/Gd@FA NCs with different properties were prepared. It can be seen from the UV spectrum of three kinds of Au/Gd@FA NCs that the absorption peak of Au/Gd@BSA NCs is located at 518 nm, and the absorption peaks of Au/Gd@BSA NCs prepared with different molar ratios did not change significantly. In addition, the modification of FA molecules had no obvious effect on the absorption peak of the probes (Figure 1A).

The fluorescence spectra of the three kinds of Au/Gd@FA NCs were significantly different from the UV spectra. From the fluorescence emission spectra, it can be seen that the maximum emission wavelengths of Au/Gd@BSA NCs prepared with different mole ratios shifted with the change of mole ratios (Figure 1B). When the excitation wavelength was 530 nm and the molar ratio of HAuCl_4_ to Gd(NO_3_)_3_ was 1:1.5, the emission wavelength of Au/Gd@BSA NCs was 690 nm, which was the longest among the three kinds of Au/Gd@BSA NCs. The Au/Gd@BSA NCs prepared with a molar ratio of 1:1 had the smallest emission wavelength of about 660 nm (Appendix A). This indicates that the change in the ratio of Au and Gd elements has an effect on the fluorescence properties of the probes when the sample concentration is constant. In addition, the fluorescence quantum yields (QY) of Au/Gd@BSA NCs prepared in three different molar ratios were also measured (Appendix A). The QY of Au/Gd@BSA NCs with a molar ratio of 1:1.5 was the highest (4.4%). Their QY was also relatively high compared to the results from other studies, showing good fluorescence properties. In order to study the fluorescence properties of Au/Gd@BSA NCs with the effect of FA conjugation, the fluorescence changes of nanoparticles before and after the attachment of FA molecules were detected. The fluorescence spectra showed that the fluorescence emission peaks of Au/Gd@FA NCs were consistent with those of Au/Gd@BSA NCs, their emission wavelengths did not change, and the fluorescence intensity significantly increased (Figure 1C). The results indicated that Au/Gd@FA NCs were successfully prepared and the fluorescence performance was effectively improved.

It is well known that in cellular and in vivo experiments, probes with long fluorescence emission wavelengths and high fluorescence intensities can effectively avoid possible biological signal interference and exhibit good signal-to-noise ratios and imaging effects. Therefore, we chose Au/Gd@FA NCs prepared in a molar ratio of 1:1.5 for subsequent experiments and performed further chemical characterization. We investigated the fluorescence stability of Au/Gd@FA NCs in different media. The results showed that the fluorescence emission wavelength of Au/Gd@FA NCs was 690 nm in water, phosphate buffer saline (PBS), and 10% fetal bovine serum (FBS) + 0.9% NaCl solution with no significant change in fluorescence intensity (Figure 1D). The fluorescence performance of Au/Gd@FA NCs was very stable and could be well applied to subsequent cellular and in vivo experiments. The particle size and morphology of Au/Gd@FA NCs were characterized by TEM. The results showed that the particle size of Au/Gd@FA NCs was about 2–3 nm (Figure 1E,F), which was ultra-small and had a uniform particle size distribution. For better use in cellular and in vivo experiments, we further tested the hydration radius of Au/Gd@FA NCs. The results of hydration radius showed that the average hydration diameter of Au/Gd@FA NCs was only 13.2 nm.

To demonstrate the success of the covalent bonding of FA to the surface of Au/Gd@BSA NCs, we measured the infrared spectrum and zeta potential of the samples. From the FT-IR spectra, it can be seen that the biggest difference in the infrared spectrum of Au/Gd@FA NCs compared to that of FA was the presence of vibrations at 1258 cm^−1^ and 1384 cm^−1^, which was attributed to covalent bonding between C-*n* [36]. This covalent bond may be formed by the condensation reaction between the -NH_2_ group on BSA and the -COOH group on FA (Figure 2A). In addition, the zeta potential changed from − 15.2 ± 0.5 mV to 0.4 ± 0.3 mV when FA was covalently attached to the surface of Au/Gd@BSA NCs (Figure 2B). This is mainly related to the -NH_2_ on the FA surface. Thus, the results of infrared spectra and zeta potential suggest that FA has been attached to the surface of Au/Gd@BSA NCs. To further determine the content of each element, the EDS data of Au/Gd@FA NCs were measured and analyzed. From the EDS spectra, both Au and Gd elements can be observed (Figure 2C). As can be seen from Appendix A, the atomic ratio of Au in Au/Gd@FA NCs is 0.12%, while the atomic ratio of Gd is 0.02%, and the atomic ratio of Au:Gd is 6:1. This proves that although the Gd content is relatively low, it has been successfully doped into Au/Gd@FA NCs.

Meanwhile, X-ray photoelectron spectroscopy (XPS) was used to further confirm the elemental composition of Au/Gd@FA NCs. The presence of Au and Gd can be detected in the full range XPS spectra of Au/Gd@FA NCs (Figure 2D). The Gd4d-specific binding energy (139 eV to 147 eV) of Au/Gd@FA NCs can be seen in Figure 2E [37]. Unfortunately, similar to the results of the EDS spectra, the signal of Gd was very weak. Therefore, the results of XPS spectroscopy also indicate that Gd has been successfully doped into Au/Gd@FA NCs at very low levels.

### 3.2. Cytotoxicity Assay and Cell Uptake

To evaluate the cytotoxicity of Au/Gd@FA NCs, tumor cells (mouse breast cancer cells, 4T1 cells) and normal cells (L02 cells) were selected for MTT assay. As shown in Figure 3A, when the concentration of Au/Gd@FA NCs was 100 μg/mL, the cell survival rate of 4T1 cells was 82.1%, which was still higher than 80%. For normal cells (L02 cells), the survival rate was higher in the same concentration range compared to 4T1 cells (Figure 3B). The experimental results showed that the survival rates of both normal cells and tumor cells were higher than 80%, indicating that Au/Gd@FA NCs had good biocompatibility and could meet biosafety requirements for subsequent in vivo experiments. In addition, tumor cells and normal cells incubated with Au/Gd@FA NCs were broken up and extracted, and the content of Gd elements was detected by ICP-MS. The results suggested that the Gd content in normal cells was about 0.17 fmol/mL, while the Gd content in tumor cells was about 0.6 fmol/mL. This indicated that the uptake of tumor cells was about 3.2 times higher than that of normal cells, and the modified FA molecules had good targeting ability. Combined with MTT assay results, it showed that Au/Gd@FA NCs had a certain targeting ability and could effectively accumulate in tumor cells, while the uptake to normal cells was low, which greatly reduced the possible toxic and side effects to normal cells.

### 3.3. Intracellular Near-Infrared FL Imaging of Au/Gd@FA NCs

Since Au/Gd@FA NCs have good near-infrared fluorescence properties, the absorption and fluorescence distribution of Au/Gd@FA NCs in tumor cells after cell culture were analyzed by confocal fluorescence microscopy to evaluate the potential of Au/Gd@FA NCs as near-infrared fluorescent contrast agents and to consider their possible application in image-guided cancer therapy. Au/Gd@FA NCs (60 μg/mL) and Au/Gd@BSA NCs (60 μg/mL) were incubated with tumor cells (4T1 cells) at 37 °C for 6 h, respectively, and the fluorescence signal was detected by confocal fluorescence microscopy (Figure 4). The results showed that the tumor cells had almost no fluorescence in the untreated control group, while the tumor cells in the Au/Gd NCs co-incubated group had only a small amount of fluorescence around the cell membrane, indicating that a large amount of Au/Gd@BSA NCs could not enter the cells effectively. Notably, the tumor cells co-incubated with Au/Gd@FA NCs showed a strong red fluorescence signal, and the fluorescence intensity was significantly higher than the other two groups, indicating that FA modified on the surface of Au/Gd@BSA NCs could better bind to the folate receptor of tumor cells and be actively taken up by tumor cells and enter the cytoplasm in large quantities. Au/Gd@FA NCs showed better near-infrared FL imaging ability. To further determine the distribution of Au/Gd@FA NCs within tumor cells (4T1 cells), we performed 3D layer-by-layer scanning of 4T1 cells co-incubated with Au/Gd@FA NCs by confocal fluorescence microscopy. The results showed that the 4T1 cells had fluorescent signals from top to bottom and that the signal intensity was stable (Appendix A). Therefore, it can be confirmed that Au/Gd@FA NCs successfully entered into tumor cells and maintained a stable fluorescence signal.

In addition, the fluorescent probes used for the assay have excellent fluorescence properties and good biocompatibility, and can also effectively distinguish tumor cells from normal cells, with accurate targeting properties. Therefore, we further investigated the selectivity of Au/Gd@FA NCs for tumor cells and normal cells. Confocal microscopic imaging results showed that the same concentration of Au/Gd@FA NCs did not observe a significant fluorescence signal in normal cells (L02), while a bright red fluorescence signal was observed in 4T1 cells (Appendix A). It was further shown that Au/Gd@FA NCs had a strong specific preference for tumor cells, and confirmed that the uptake of Au/Gd@FA NCs by tumor cells could be significantly increased by modifying FA molecules, and could accumulate in the cytoplasm. This result was confirmed by ICP-MS.

### 3.4. In Vivo Near-Infrared FL Imaging Metabolism of Au/Gd@FA NCs

An in vitro FL imaging study confirmed that Au/Gd@FA NCs can maintain good near-infrared fluorescence properties in cells, can effectively identify tumor cells, and accumulate in large quantities in tumor cells. Inspired by the above study, 4T1 tumor-bearing mice were used as animal models to observe the in vivo FL imaging effect of Au/Gd@FA NCs (100 μg/mL, 200 μL) after tail vein injection. After 2 h of intravenous injection, a fluorescence signal was clearly observed and accumulated at the site of the tumor, while no fluorescence signal was detected in the surrounding normal tissues. Over time, the fluorescence signal of the tumors gradually increased and reached a maximum 8 h after injection. The fluorescence signal could completely outline the complete morphology of the tumor, and the fluorescence intensity did not decay within 2 h, which started to gradually decrease from 8 h after injection to 10 h, and then began to decrease gradually after injection 10 h (Figure 5A). The in vivo study once again demonstrated that Au/Gd@FA NCs had good near-infrared fluorescence properties and targeting, which could be effectively accumulated in tumors through blood circulation for efficient fluorescence imaging of solid tumors. To further explore the distribution of Au/Gd@FA NCs in mice, the mice were sacrificed after 8 h of injection, when the tumor fluorescence signal was strongest. Major organs and tumors were collected and observed using the FL imaging system. As shown in Figure 5B, the fluorescence images of ex vivo organs showed the strongest fluorescence signal within the tumor, a small amount of signal in the liver, and no significant fluorescence signal in other major metabolic organs. This result indicated that Au/Gd@FA NCs had excellent targeting and effectively accumulated in solid tumors, which validated each other with the fluorescence imaging results.

Notably, in the in vivo FL imaging study, we found that the fluorescence signal at the tumor site completely disappeared after intravenous injection of Au/Gd@FA NCs for 36 h, but the fluorescence signal in the intraperitoneal cavity was significantly enhanced, and an obvious fluorescence signal was detected in feces (Figure 5A). This phenomenon suggested that Au/Gd@FA NCs may be excreted through the feces of mice. To verify the metabolism of Au/Gd@FA NCs in mice, the main organs and tumors of mice were collected after 36 h of injection. Fluorescence imaging of the organs ex vivo was analyzed. The results showed that a strong fluorescence signal was detected in the intestinal (Figure 5C), indicating that Au/Gd@FA NCs may be excreted through the liver-fecal pathway in mice. Previous studies have shown that there are two main pathways for the excretion of nanomaterials in mice: the “renal-urinary” pathway and the “hepatic-fecal” pathway. The present study reconfirmed this conclusion [38,39]. The small particle size of Au/Gd@FA NCs, with a hydration radius of 13.2 nm, can be rapidly excreted through renal metabolism. Meanwhile, this study found that Au/Gd@FA NCs can also be excreted into the intestine to form feces through hepatic metabolism. The combination of the two metabolic pathways improved the metabolism of Au/Gd@FA NCs, effectively reducing the amount of residues in vivo, and improving biosafety. This was corroborated by the absence of significant abnormalities in the diet, water intake, mental status, and behavioral characteristics of mice throughout the in vivo experiments.

Therefore, the results of the in vivo FL imaging study showed that Au/Gd@FA NCs still maintained stable fluorescence performance and good targeting after injection into mice, and effectively concentrated on the tumor site through blood circulation in vivo to achieve accurate FL imaging of tumors with good stability. Meanwhile, Au/Gd@FA NCs can be maximally eliminated from the body through two metabolic pathways: renal metabolism and hepatic metabolism, which greatly reduces the residual nanomaterials in the body and effectively protects the normal physiological metabolism of various organs in the body.

### 3.5. In Vivo T_1_-Weighted MRI of Au/Gd@FA NCs

Au/Gd@FA NCs have successfully achieved accurate near-infrared FL imaging of tumors at the cellular and in vivo levels due to their stable near-infrared fluorescence properties and good targeting. In addition, Au/Gd@FA NCs contain a small amount of Gd and have good paramagnetic properties for MRI. Therefore, we explored the possibility of Au/Gd@FA NCs as MRI contrast agents. First, we compared the nuclear magnetic properties of the prepared Au/Gd@FA NCs with those of a commercial MRI contrast agent, Gd-diethylenetriamine pentaacetic gadolinium (Gd-DTPA), to evaluate its potential application in MRI. As can be seen from the T_1_-weighted MRI images in Figure 6A,B, the signal intensity is still much higher than that of the high concentration Gd-DTPA contrast agent even at a low concentration of Au/Gd@FA NCs [40]. The longitudinal proton relaxation time T_1_ was measured and the corresponding Gd^3+^ concentration in Au/Gd@FA NCs probes was measured by ICP-MS. The relaxation value *r*_1_ was determined by plotting these two values with a function. The results showed that the *r*_1_ value of Au/Gd@FA NCs was 14.65 mM^−1^ S^−1^ (Gd) at 1.5 T, which was 3-fold higher than that of Gd-DTPA (Figure 6C). Furthermore, it can be concluded that Au/Gd@FA NCs have excellent T_1_ imaging ability. The excellent performance of Au/Gd@FA NCs as MRI contrast agents in vitro prompted us to further explore their feasibility in vivo. After 200 μL Au/Gd@FA NCs (100 μg/mL) were injected intravenously into tumor-bearing mice for 8 h, the nude mice were examined by T_1_-weighted MRI. As shown in Figure 6D, on the T_1_-weighted MRI images, it was observed that the brightness of the tumor site was significantly higher than that of other normal tissues, which could effectively distinguish the lesion site and achieve MRI images of the tumor. The results confirm that Au/Gd@FA NCs can be used as a potential T_1_-weighted MRI contrast agent.

The results of near-infrared FL imaging and T_1_-weighted MRI showed that the prepared Au/Gd@FA NCs had good near-infrared fluorescence properties and a strong T_1_ effect. In the presence of FA, Au/Gd@FA NCs actively accumulated effectively in the tumor and achieved near-infrared FL imaging and T_1_-weighted MRI at the tumor site. Meanwhile, combined with metabolic analysis, the residual amount of Au/Gd@FA NCs in vivo is small and has excellent biosafety. Combined with the fluorescence detection results, Au/Gd@FA NCs are expected to be metabolizable dual-mode contrast agents for near-infrared FL imaging and T_1_-weighted MRI.

### 3.6. Intracellular ROS Detection and Analysis by Laser Irradiation

In vivo FL imaging and MRI studies, Au/Gd@FA NCs have been found to be actively enriched at the tumor site. Due to the strong SPR absorption band of gold-based nanoparticles, Au/Gd@FA NCs enriched in large amounts at tumor sites are highly susceptible to heat generation under light due to the strong SPR effect, which affects the physiological metabolism of tumor cells. Under laser irradiation at 660 nm (1.2 W/cm^2^), the temperature of Au/Gd@FA NCs solution (60 μg/mL) increased from 21 °C to 46.1 °C within 6 min, while PBS as the control group only increased to 35.4 °C (Figure 7A). The results indicate that Au/Gd@FA NCs solution can convert light energy into heat energy under laser irradiation and has some photothermal conversion ability. In addition, we investigated the changes of ROS in tumor cells after co-incubation with Au/Gd@FA NCs under laser irradiation. The generation of ROS in cells was detected using the sensitive fluorescent dye DCFH-DA, and the fluorescence intensity was proportional to the generation of ROS. Au/Gd@FA NCs (60 μg/mL) were co-incubated with 4T1 cells for 8 h, and then DCFH-DA solution was added for further incubation for 45 min, and the fluorescence intensity was observed under the confocal fluorescence microscope. As shown in Figure 7B, after 660 nm (1.2 W/cm^2^) laser irradiation for 6 min, the 4T1 cells showed clear green fluorescence, indicating that a large amount of ROS was produced in the cells. However, there was only a weak fluorescence signal in the non-irradiated group, indicating that the ROS content in the cells was low. The fluorescence intensity in the laser irradiation group was 5.3 times higher than that of the non-irradiated group (Figure 7C). This indicated that Au/Gd@FA NCs could generate a large amount of ROS in tumor cells under light irradiation, which might be due to the enhanced interaction between bipolar FA molecules and plasma field Au/Gd@BSA NCs [41].

Therefore, we found that the prepared Au/Gd@FA NCs had some photothermal conversion ability and could promote ROS production in tumor cells. Photothermal effects enhance tumor cells, with significant killing effects, while photogenerated intracellular ROS (such as singlet oxygen) can increase the biological damage of exposed cells through gold-enhanced singlet oxygen photogeneration [42,43]. To further clarify the influence of Au/Gd@FA NCs entering tumor cells under laser irradiation to produce a large number of ROS, leading to the death of tumor cells. We first used MTT to investigate the toxicity of tumor cells under the same laser irradiation conditions and compared with the non-laser irradiation group. MTT assays were performed on 4T1 cells co-incubated with different concentrations of Au/Gd@FA NCs solution under 660 nm laser irradiation (1.2 W/cm^2^) and non-irradiation survival rate. Compared with the non-irradiated group, the cell survival rate of the laser-irradiated group was significantly decreased, and at 100 μg/mL, the cell survival rate was only 26% (Figure 7D), indicating that Au/Gd@FA NCs could generate a large amount of ROS and photothermal effects to kill tumor cells under laser irradiation. The YO-PRO-1/PI apoptosis kit can be used to stain apoptotic and necrotic cells. The apoptotic cells are dyed green fluorescence by YO-PRO-1, and necrotic cells can be stained with YO-PRO-1 and PI at the same time. The superposition of red fluorescence and green fluorescence shows orange yellow. The results showed that after 6 min of 660 nm (1.2 W/cm^2^) laser irradiation, many 4T1 cells were stained orange, indicating that the cells were obviously necrotic (Appendix A). Confocal experiment results were consistent with MTT results, indicating once again that Au/Gd@FA NCs entered tumor cells and produced a large number of ROS, which destroyed the physiological metabo-lism of cells and led to a large number of cell deaths. Inspired by the above research results, Au/Gd@FA NCs synergize mild photothermal effects and intracellular ROS production under laser irradiation and are expected to be one of the candidate materials for tumor-targeted therapy.

## 4. Conclusions

In this study, Au/Gd@BSA NCs were prepared by a simple hydrothermal synthesis method, and FA was modified on their surface to prepare Au/Gd@FA NCs with near-infrared fluorescence properties and strong T1 effect to achieve FL/MRI-targeted dual-mode imaging of tumors. Meanwhile, Au/Gd@FA NCs have a photoresponsive effect, which can produce a large amount of ROS and photothermal effects in tumor cells under laser irradiation and have a good killing effect on tumor cells. FA molecule modification promotes enrichment of Au/Gd@FA NCs in tumor cells, improves fluorescence quantum yield, realizes site-specific targeting, and improves FL imaging accuracy and precision therapy efficiency. Under laser irradiation, Au/Gd@FA NCs can promote ROS production and photothermal conversion in tumor cells due to the interaction between FA and Au/Gd@BSA NCs, and synergistically kill tumor cells. In addition, the smaller particle size Au/Gd@FA NCs can be metabolized by the kidney and liver and excreted by urine and feces with good biosafety. The collective nature of Au/Gd@FA NCs makes it a tumor-targeted bimodal imaging and an excellent photosensitizing therapeutic agent, providing a new approach for the development of metabolizable clinical multimodal imaging and efficient tumor therapy.

## Data Availability

Not applicable.

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
