# Peer review of "Ultra-Small and Metabolizable Near-Infrared Au/Gd Nanoclusters for Targeted FL/MRI Imaging and Cancer Theranostics"

_biosensors, 2022, doi:10.3390/bios12080558_

Round 1

Reviewer 1 Report

This study is original and well designed, however one experiment is missing. There is no proof at this stage that this is a theranostic agent, since the effect of the tumor ROS production have not been proven (i.e. measurement of tumor apoptosis via diffusion MRI or ex vivo via IHC). At this stage, it is qualified as a diagnostic agent and one additional experiment is needed to qualify it as a theranostic agent.

Line 275 why does a threshold of 80% guarantee biosafety?

Line 312 does a selectivity of 3 constitute an accurate targeting?

Line 361 if the hepatic fecal pathway is involved in the metabolization of these nanoparticles, why is the signal in the liver so weak with respect to the intestine? Did you repeat the experiment on additional mice?

Line 374 You suggest the renal pathway is also involved: how come there is no signal in the kidneys?

Line 368 can we really talk about a mental status with respect to mice? How do you assess this parameter?

Line 436 does the increase in the amount of ROS in the tumor tissue result in a change in tumor cell density? this could be achieved using diffusion MRI for example. At this stage, there is no proof of a real theranostic agent (therapy efficiency is not proven). The conclusion of the abstract is currently overstated. No apoptosis experiments have been performed.

Line 475 please discuss about the translational properties of the method in details.

Reviewer 2 Report

In this manuscript, the authors designed a hybrid nanomaterial on detecting solid tumor by fluorescence imaging and MRI. Here are some suggestions and questions:

1The procedure of synthesis in scheme 1 could not help understand what they have designed. All the reagents including Gd(NO3)3·6H2O, Gold chloride acid, BSA and FA were not presented in the products. I hope they could draw a new one to explain the procedure more clearly.

2FA-NH2 mentioned in 2.1 was absent on Aladdin’s website, and I also couldn’t make sure what FA-NH2 exactly is, please check its name and make it standardized.

3The procedure about one-step hydrothermal described in 3.1 was not matched to that in 2.2. And the reaction condition of 37was not matched to the definition of hydrothermal.

4About the increasing of fluorescence intensity after FA modified, I think a more reasonable explanation is needed. And actually, they didn’t have direct evidence to prove that FA was successfully modified on NCs. Because the changing on fluorescence intensity and zeta potential could also be considered that the procedure adding FA made BSA detaching from NCs. Therefore, there should be more evidences focusing on proving FA being coated on the NCs rather than the changes after modifying.

5Since FA was linked by its -COOH combining with -NH2 on BSA, why did they use FA-NH2 rather than FA monocase?

6From line 264 to 267, Unfortunately, similar to the results of the EDS spectra, the signal of Gd is very weak. However, the results of XPS spectroscopy also indicate that Gd has been successfully doped into Au/Gd@FA NCs at very low levels. The results of XPS and EDS were supportive for the same thing, which is Gd is positive but low content, then what is the word However for?

7They characterized cell uptaking by breaking up and extracting the cells incubated with NCs and detecting the content of Gd. I don’t think it’s persuasive because this could not exclude the situation of NCs attaching on the cell surface. The result of confocal in Figure 4 could have proved that, but the images clarity was a little poor, and lack of scale bar. And why did they put the title cell uptake in cytotoxicity rather than in vitro fluorescence?

8All the in vivo experiments were lack of a control group of Au/Gd@BSA NCs, which was important to show the necessity of modifying FA.

Round 2

Reviewer 1 Report

All comments have been properly addressed. The paper is now acceptable as such. Kind Regards